# Laser Powder Bed Fusion of Inconel 718 Tools for Cold Deep Drawing Applications: Optimization of Printing and Post-Processing Parameters

**DOI:** 10.3390/ma16134707

**Published:** 2023-06-29

**Authors:** Cho-Pei Jiang, Andi Ard Maidhah, Shun-Hsien Wang, Yuh-Ru Wang, Tim Pasang, Maziar Ramezani

**Affiliations:** 1Department of Mechanical Engineering, National Taipei University of Technology, Taipei 10608, Taiwan; 2High-Value Biomaterials Research and Commercialization Center, National Taipei University of Technology, Taipei 10608, Taiwan; 3College of Mechanical and Electrical Engineering, National Taipei University of Technology, Taipei 10608, Taiwan; ardmaidhah13@gmail.com; 4Graduate Institute of Mechatronics, National Taipei University of Technology, Taipei 10608, Taiwan; 5Department of Materials and Mineral Resources Engineering, National Taipei University of Technology, Taipei 10608, Taiwan; yavis6509wang@gmail.com; 6Department of Manufacturing and Mechanical Engineering Technology, Oregon Institute of Technology, 3201 Campus Drive, Klamath Falls, OR 97601, USA; tim.pasang@oit.edu; 7Department of Mechanical Engineering, Auckland University of Technology, 55 Wellesley Street East, Auckland 1010, New Zealand; maziar.ramezani@aut.ac.nz

**Keywords:** power bed fusion, Inconel 718, deep drawing, optimization, post-processing

## Abstract

Inconel 718 (IN 718) powder is used for a laser powder bed fusion (LPBF) printer, but the mechanical properties of the as-built object are not suited to cold deep drawing applications. This study uses the Taguchi method to design experimental groups to determine the effect of various factors on the mechanical properties of as-built objects produced using an LPBF printer. The optimal printing parameters are defined using the result for the factor response to produce an as-built object with the greatest ultimate tensile strength (UTS), and this is used to produce a specimen for post-processing, including heat treatment (HT) and surface finishing. The HT parameter value that gives the maximum UTS is the optimal HT parameter. The optimal printing and HT parameter values are used to manufacture a die and a punch to verify the suitability of the manufactured tool for deep drawing applications. The experimental results show that the greatest UTS is 1091.33 MPa. The optimal printing parameters include a laser power of 190 W, a scanning speed of 600 mm/s, a hatch space of 0.105 mm and a layer thickness of 40 μm, which give a UTS of 1122.88 MPa. The UTS for the post-processed specimen increases to 1511.9 MPa. The optimal parameter values for HT are heating to 720 °C and maintaining this temperature for 8 h, decreasing the temperature to 620 °C and maintaining this temperature for 8 h, and cooling to room temperature in the furnace. Surface finishing increases the hardness to HRC 55. Tools, including a punch and a die, are manufactured using these optimized parameter values. The deep drawing experiment demonstrates that the manufactured tools that are produced using these values form a round cup of Aluminum alloy 6061. The parameter values that are defined can be used to manufacture IN 718 tools with a UTS of more than 1500 MPa and a hardness of more than 50 HRC, so these tools are suited to cold deep drawing specifications.

## 1. Introduction

Additive manufacturing (AM) is a layer-stacking technology that forms material, layer by layer, into a three-dimensional physical object. AM is used to manufacture complex geometrical objects with low material waste and produces objects faster than a subtractive manufacturing method. AM is used in the biomedical, automotive, aerospace and tooling industries. Powder bed fusion (PBF), sheet lamination (SL), binder jetting (BJ), material extrusion (ME), and direct energy deposition (DED) are techniques to print metal parts. PBF is the most common method because it is financially expedient, and production time is reduced. It also produces metal objects with acceptable accuracy and with mechanical properties that are similar to those of parts that are produced using conventional manufacturing methods [1,2,3,4].

Laser powder bed fusion (LPBF) is used to print metal objects with complex geometries. LPBF uses a high-intensity laser to melt metal powder using a predefined scanning path for each layer, so layers are stacked to form a three-dimensional (3D) object. The printing parameters include laser power, scanning speed, hatch space, and layer thickness, and these are defined before the printing process is conducted [5,6,7]. These four parameters affect the relative density, surface roughness, and mechanical properties of an as-built object that is produced using an LPBF printer.

Inconel (IN) 718 powder is a nickel-based super-alloy that is used for LPBF printers [1,6,7,8,9]. It features high wear resistance, superior corrosion resistance, and excellent mechanical properties that remain stable at high temperatures. It is used to manufacture turbine blades for the power industry, jet turbines for the aerospace industry, and to manufacture mould and die parts for metal forming because mechanical properties are improved [10,11]. The results of several studies on the printing parameters for IN 718 are summarized in Table 1, which shows that different LPBF printers use different printing parameters. The values for printing parameters must be optimized to maximize accuracy and to optimize the good mechanical properties of an as-built object. The optimization of printing parameters by trial-and-error wastes time and money.

The optimization of parameter values for a material for an LPBF printer requires a long research process [19,20]. The Taguchi method can be used to optimize the printing parameter values for an LPBF printer for specific materials. The time and material cost [21] for an experiment is reduced because a design of experiment replaces trial and error [22]. One study’s results for the printing parameters for IN 625 for an LPBF show that the laser power most significantly affects hardness and surface roughness, followed by the scanning speed and hatch space [23]. Another study determined the effect of laser power, scan speed, and hatch space on the micro-hardness and surface roughness of printed IN 625 samples, and the results show that the optimal values for printing parameters are a laser power of 270 W, a scan speed of 800 mm/s and a hatch space of 0.08 mm, which produce a micro-hardness of 416 HV and a surface roughness of 2.82 μm [21]. Analysis of variance (ANOVA) is used in many of the Taguchi methods as a test procedure to solve the problem of optimizing parameters for an output in a process [24]. SS 316L was fabricated with LPBF using ANOVA as a test procedure, verifying the effect of printing parameters in reducing porosity formed in the fabrication process. Their study result shows a 74.9% reduction in porosity [25]. However, no studies use the Taguchi method to optimize the values for printing parameters for IN 718 or to determine the factors that affect the ultimate tensile strength of the as-built object.

AM is used for the direct printing of metal dies [26]. LPBF was also used to print die-casting inserts, and the mechanical properties of the printed inserts render them unsuited for use in the production of a die. Printed insert dies must be post-processed to improve the mechanical properties of the as-built object [27]. Cold deep drawing is one type of metal formed in the manufacturing process. This cold working process can produce metal parts at temperatures from room temperature to 30% of the melting point of the material being worked. The mechanical properties of a cold drawing die include an ultimate tensile strength (UTS) of more than 1500 MPa and a hardness of more than 50 HRC [28,29].

Another study printed a die using H13 for hot extrusion using an LPBF printer, and the experimental results show that there are high residual stresses in the as-built H13 die, so it is unsuitable for direct extrusion because high residual stress leads to cracking easily when the material deforms in the extrusion die [30]. This shows that the mechanical properties of as-built metal die that are produced using an LPBF printer must be measured to determine their suitability for die applications. Post-processing improves the mechanical properties of the as-built object.

Heat treatment (HT) is used in the manufacturing industry to improve the mechanical properties of metals. IN 718 can undergo heat treatment using precipitate hardening. Generally, precipitate hardening has two steps: the first is solution treatment to produce the γ phase, and the second is artificial ageing to produce the γ′ and γ″ phases [31,32]. However, some research was performed for double artificial ageing to improve the mechanical properties of IN 718. Double ageing (DA) was used for IN 718 treatment with temperature of the specimen being 700 °C, with a holding time of 12 h for the first ageing sequence. The temperature is then reduced to 620 °C, with a hold time of 6 h as the second ageing sequence, followed by air cooling. DA increases the UTS for IN 718 to more than 1500 MPa [26,33]. IN 718 manufactured by directed energy deposition has shown the best performance of creep phenomenon after DA treatment. Its creep lifetime is 200/h, the highest value compared to other heat treatments (HT), such as HT homogenization and hot isotactic pressing and DA [34]. However, some studies explained that, without solution treatment as the first treatment in precipitation hardening of IN 718, this made the formed γ phase unstable [32].

This study fabricates IN 718 tools for a round cup deep drawing application using an LPBF printer. However, as far as the author knows, there is still no study that applies LPBF to print parts for cold deep drawing. In addition, the Taguchi method is used to determine the optimal printing parameter values for IN 718 powder that produce the best mechanical properties in terms of UTS, hardness, and surface roughness. The study then determines whether the optimal parameter values for the DA treatment for as-built IN 718 specimens produce mechanical properties that are suited to a deep drawing application. Tools are produced and set up on a stamping machine to verify their suitability in a round cup deep drawing application.

## 2. Materials and Methods

### 2.1. Research Flow

This study proposes a standard procedure to print IN 718 tools for a cold deep drawing application. The mechanical properties of the as-built, heat-treated, and surface-finished objects are determined. This study used an LPBF printer and the Taguchi method to optimize the printing parameter values for IN718 that produce the greatest UTS. These optimized printing parameters are then used to manufacture a specimen for DA treatment using different parameter values. The UTS of the printed die must be greater than 1500 MPa, and the hardness must be greater than 50 HRC for a cold deep drawing application.

Figure 1 shows the research flow for this study. Previous studies show that four factors affect the UTS: laser power, scanning speed, hatch space, and layer thickness. The printing parameter values for 9 experimental groups were determined using the Taguchi method. The specimens for each group were printed, and a tensile test was conducted. A comparison of the UTS values for the 9 groups shows the printing parameter that has the maximal UTS, and this is optimized using the result of the Taguchi method. The optimized printing parameter is used to reprint the tensile specimens. Half of the tensile specimens underwent DA treatment using different parameter values, and all specimens were subjected to a tensile test to determine the effect of DA treatment and the surface finish on the UTS and hardness. The optimal printing and DA parameter values were then used to produce tools for a deep drawing test.

### 2.2. Material Preparation and Laser Powder Bed Fusion Printer

IN 718 powders were purchased from Chung Yo Materials Co., Ltd., Kaohsiung, Taiwan. These were subject to sieving and drying before the printing process. The powder was sieved using a filter with a mesh size of 60 µm. The drying process used a temperature of 150 °C and a holding time of 1 h. Figure 2 shows a scanning electron microscope (SEM) image of IN 718 powder, which has a diameter of 10–50 µm.

An LPBF printer (AMP-160, Tongtai Co., Kaohsiung, Taiwan), which uses selective laser melting, was used to print the specimen. The specimen model was exported using a standard tessellation language (STL) format and imported to a slicing software (Materialise Magics 23.1, Lauven, Belgia) to generate the scanning path for the printing parameter values.

### 2.3. Taguchi Optimization for the Printing Parameter

The Taguchi method was used to design the experiments and to determine the relationship between the printing parameters and their effect on the mechanical properties of as-built objects, including UTS and hardness. The printing parameters are laser power (*P*), scan speed (*V*), hatch spacing (*H*) and layer thickness (*Lt*), which are calculated using the volumetric laser density (VED) formula (J/mm^3^) as Equation (1):(1)$$VED=\frac{P}{V\cdot H\cdot Lt}$$

This study uses the UTS as the criterion for the Taguchi method. The greater the UTS, the more optimized is the printing parameter. The signal-to-noise (*S*/*N*) ratio is a standard for quality control and is expressed as Equation (2). Increasing the *S*/*N* value decreases the standard deviation, so the parameters are more stable [19,35].
(2)$$\frac{S}{N_{i}}=-10\;log\left(\frac{1}{n}\sum_{i=1}^{n}\frac{1}{y_{i}^{2}}\right)$$

The printed tensile specimens were subjected to tensile testing. The specimen details pertain to ASTM E8 standards. Before the printing parameters were optimized, the tensile specimen was printed using the vertical and horizontal building, as shown in Figure 3, to determine the effect of building direction difference on the tensile test result. The best build direction was used for the experiment.

The printing parameter values in Table 2 show that the minimum and maximum values for laser power, scanning speed, and layer thickness are 180 W and 200 W, 600 mm/s and 800 mm/s, 0.08 mm and 0.105 mm, and 30 microns and 50 microns, respectively. The Taguchi method for this study uses a level of 3, and the printing parameter table is shown in Table 2. An orthogonal *L*_9_ table is created using Table 2 and is shown in Table 3. Each parameter in the orthogonal table occurs the same number of times, so an analysis of variance (ANOVA) is used to determine the effect of each parameter on the printing quality. A lab-developed software, based on ANOVA, was used to create the response factor plots of the average of UTS and the *S*/*N* ratio.

Each set of experiments was repeated 6 times (r = 6), and there were nine sets of experiments, generating 74 data sets. The experimental results are expressed in terms of the mean value (calculated using Equation (3)), standard deviation (calculated using Equation (4)), and *S*/*N* value (calculated using Equation (2)).
(3)$$\bar{y_{i}}=\sum_{j=1}^{r}\frac{y_{ij}}{r}$$
(4)$$S_{i}=\sqrt{\frac{\sum_{j=1}^{r}{\left(y_{ij}-\bar{y_{i}}\right)}^{2}}{r-1}}$$
where; $y_{ij}$ refers to the experimental data, i is the experiment of the sample code, and j is the result of the r^th^ investigation.

### 2.4. Heat Treatment

IN 718 can be strengthened through precipitation hardening, and one of the types is double ageing (DA). In this study, the as-built IN 718 specimens involved DA using a furnace (HTF 1800, Carbolite Gero, Derbyshire, UK). The first ageing was carried out at a temperature of 720 °C and the second at 620 °C. The holding time in this study was varied to obtain maximum tensile strength. First ageing has holding time variations of 6, 8, and 10 h. Meanwhile, second ageing has various holding times of 8, 10, and 12 h. For more details, results can be observed in Table 4.

IN 718, printed with LPBF, according to the optimized printing parameters, was heated in the furnace until it reached 720 °C and held for a specific duration. After the holding process, the material was slowly cooled in the furnace at 55 °C per hour. The second ageing process was reheating from room temperature to 620 °C and involved holding the material at a specific time. After the holding time process, the material was cooled in the air until it reached room temperature.

### 2.5. Hardness and Surface Roughness Measurement

Hardness was measured for the as-built, heat-treated, and surface-finished samples. Surface finishing used shot peening and polishing. The optimized printing parameter values were used to print the specimen. The hardness test used the Vickers method with ASTM E384 as the testing material standard. The hardness was measured using a Vickers hardness testing machine (HMV-G21S, Shimadzu, Kyoto, Japan). Meanwhile, surface roughness testing used visual observation with the standard of ASTM D7127. The surface roughness was measured using a three-dimensional microscope (VR300 Keyence, Osaka, Japan) with the unit of µm for the roughness average (Ra).

Figure 4a shows the die for cold deep drawing that is produced by this study. Most dies have round corners to allow the sheet material to flow into the die cavity easily, so the hardness and surface roughness are measured on the upper side, the curved side, and the edge of the die. Figure 4b shows the as-built specimens.

### 2.6. Deep Drawing Application

A die and a punch were produced for a cup deep drawing experiment using the optimized printing and DA parameter values in order to determine the suitability of the material to a deep cold drawing application. A simple round cup shape was used to eliminate complication in the validation of the process, as shown in Figure 5a. The post-processed die was assembled using the blank holder, positioning pins, and a bottom plate. The assembled die set was fixed on a mounting table. The post-processed punch was fixed to the punch holder and driven using a press ram. An aluminum alloy (Al) 6061 sheet with a thickness of 1 mm was placed between the post-processed die and the blank holder. The press ram drove the punch downward to force the sheet to flow into the die to form a round cup.

Figure 5b shows the dimensions of the die and punch. The die is 95 × 95 × 9 mm^3^. The center of the die has a hole with a diameter of 39.42 mm. The diameter and height of punch are 37.42 mm and 33 mm, respectively. The edges of the hole on the top side and the punch on the contact side have a radius of 3 mm. The thickness axis is the building direction for the LPBF printer.

## 3. Results and Discussion

### 3.1. Processing Parameters Optimization by the Taguchi Method

Specimens were printed in a horizontal and vertical orientation and were subjected to a tensile test, as shown in Figure 6. The results of the tensile test and the *S*/*N* ratio are listed in Table 5. The average ultimate strength (UTS) values for the specimens that are printed using a vertical building direction are less than those that are printed using a horizontal building direction. The *S*/*N* ratio for the horizontally printed specimens is greater than 60, and the smallest average UTS is 1046.31 MPa. The printing optimization and heat treatment tests use horizontally printed specimens. This result is supported by previous research [36] as a material printed with PBLF, which has a higher tensile strength for the building direction of horizontal, rather than vertical.

To optimize the printing parameters, the response graphs for average UTS and *S*/*N* ratio are plotted in Figure 7. Laser power, scanning speed, hatch space, and layer thickness are, respectively, denoted as A, B, C, and D. Figure 7a shows that the order in which the response factors affect the UTS results for printed IN718 is: layer thickness, hatch space, scanning speed, and laser power. This result contrasts with the results of another study [23] because each study uses a different printing mechanism. The previous study used a six-axis robot with a fiber laser and a powder feeder system to deposit the powder on the laser focus zone. The printing mechanism for this study paves the powder on the platform and deposits the laser energy selectively along the scanning path to induce a phase change in the powder from the solid state to the liquid state.

The *S*/*N* ratio plot in Figure 7b shows that an increase in the hatch spacing (C) and layer thickness (D) increases the UTS for the as-built object. The powder has an average particle size of 10–50 μm, so the layer must be thicker than 40 μm. The hatch space for Sample 4 is increased to 0.105 mm from 0.09 mm, and this is denoted as Sample 10. For 6 test pieces that were printed using a horizontal building direction, the highest UTS value is 1122.88 MPa. The optimal values for the printing parameters are listed in Table 6, and the volumetric laser density formula is 75.4 J/mm^3^. The greatest value for UTS is less than 1500 MPa [26], so heat treatment is required. The as-built specimens for this experiment use the printing parameters for Sample 10.

### 3.2. Heat Treatment

The minimum respective values for UTS and hardness for cold deep drawing tools are 1500 MPa and 50 HRC [28,29]. The maximum UTS value for Sample 10 is 1122.88 MPa, which is less than 1500 MPa, so DA treatment was used to increase the UTS for the as-built object using the design in Table 4. The UTS after DA treatment is listed in Table 4 and shown in the Figure 8. The experimental results show that the UTS for the T1, T2, T3, and T4 groups is significantly greater than 1500 MPa, and T4 features the highest UTS. Therefore, the optimal DA treatment involves heating the printed specimens to 720 °C at 10 °C per minute and maintaining this temperature for 8 h and then cooling in the furnace at a cooling rate of 55 °C per hour to a temperature of 620 °C and maintaining this temperature for 8 h.

Based on Figure 8, T5 shows the lowest UTS of IN 718 value while the duration of the first and second artificial ageing holding time is neither the shortest nor the longest duration. This process is due to the unstable γ phase in multiple ageing. Some elements that cannot be released during the γ formation process, such as niobium, titanium, and molybdenum, result in other phases not being formed stably or even not being formed. This has an impact on reducing the strength of IN 718 [31].

### 3.3. Surface Finishing and Hardness Measurement

Figure 9 shows the results for surface roughness. The surface roughness of the as-built specimen is greatest on the upper side, which has a value of 11.72 µm for the Ra, but the upper side becomes slightly smoother after DA treatment. The surface roughness of the curved side is greatest because it must have a staircase effect, but experimental results show that it is less than the value for the upper side, possibly because the layer is thinner, so the staircase effect is eliminated [37]. After DA treatment and surface finishing, the surface roughness is reduced to 2 µm for the Ra.

Figure 10 shows the Rockwell hardness results. Figure 10a shows that the distance between each indentation point on the test object is 4 mm in order to avoid measurement errors when the material changes phase. Figure 10b shows that the respective hardness of the as-built specimens, specimens that are subject to DA, and post-processed specimens, which include HRC 32, HRC 46, and HRC 55.

Amato et al. annealed an as-built specimen at 982 °C for 0.5 h under vacuum and then used a hot isostatic pressing process (HIP) at 1163 °C and 0.1 GPa pressure for 4 h in argon. The maximum hardness of the as-built and annealed with HIP specimens is 33 HRC and 38 HRC, respectively [38]. Compared to it, the hardness of the treated specimen for this study is greater than 38 HRC and increases to 55 HRC after surface finishing. This reveals that the optimal parameter of this study can approach to practical application.

### 3.4. Deep Drawing Verification

Figure 11a and Figure 11b, respectively, show the as-built die and punch on the platform. Both were subject to DA treatment and were surface polished using wire-electrical discharge machining (WEDM) after they were removed from the platform. The dimensions of both comply with the specifications, and they were assembled on the punch holder and die set, as shown in Figure 11c,d. Figure 12 (left) shows a sheet on the die that is clamped using the blank holder. After deep drawing, a round cup was formed, as shown in Figure 12 (right). The shape of the round cup is simple, but it verifies that the optimized printing and DA treatment parameter values produce a punch and die that are suited for use in a cold deep drawing application.

This study optimizes the printing parameter values for IN 718 and the post-processing parameter values for the as-built parts to create mechanical properties that render this material suited to the production of a cold deep drawing die, but service life for the round cup deep drawing process is not considered, so future studies will optimize the topological structure of as-built die to eliminate material waste and increase the service life of the post-processed die.

## 4. Conclusions

This study optimizes the values for the printing and heat treatment parameters for IN 718 to produce the tools that are suited for use in a cold deep drawing application. The Taguchi method is used to determine the effect of the printing parameters on the mechanical properties of the as-built object. The following conclusions are drawn:The results of the Taguchi method show that the order in which the relevant factors affect LPBF printing is: layer thickness, hatch space, scanning speed, and laser power. However, changing the value for hatch space has the most significant effect because the diameter of the powder particles defines the least thickness for each layer.The optimized printing parameter values include a laser power of 190 W, a scanning speed of 600 mm/s, a hatch space of 0.105 mm, and a layer thickness of 40 μm to produce a maximum UTS of 1122.88 MPa. The hardness of the as-built specimen is 32.33 HRC.The optimal parameters for heat treatment are a temperature of 720 °C with a holding time of 8 h for the first ageing sequence, a decrease in temperature to 620 °C with a holding time of 8 h for the second sequence, and cooling in the furnace at a cooling rate of 55 °C per hour. After heat treatment, the UTS increases to 1511.9 MPa, and the hardness increases to 46.06 HRC. After surface finishing, the hardness increases to 55.37 HRC.The optimized values for the printing and heat treatment parameters give a tensile strength of more than 1500 MPa and a hardness of more than 50 HRC, which meet the requirements for a tool for a cold deep drawing application.The results of the deep drawing experiment verify that the optimized values for the printing and post-processing parameters produce a die and punch that form an Al 6061 round cup.Although the process of optimizing printing parameters and double aging produces materials that are suitable for standard die parts for cold deep drawing, judging from the results of the Taguchi calculations, the results are in the unsatisfactory category and can be further improved with advanced Taguchi analysis.In terms of material, there is still much that can be explored for the application of IN 718, which is printed using LPBF as a cold deep drawing dies part, such as post-printing material characterization, mechanical behavior, fatigue and failure behavior, and many others. Optimizing printing parameters using other parameters is still very possible to do in the future. This is because the printing parameters are not only related to the four parameters that we mention in this study.

## Figures and Tables

**Figure 1 materials-16-04707-f001:**
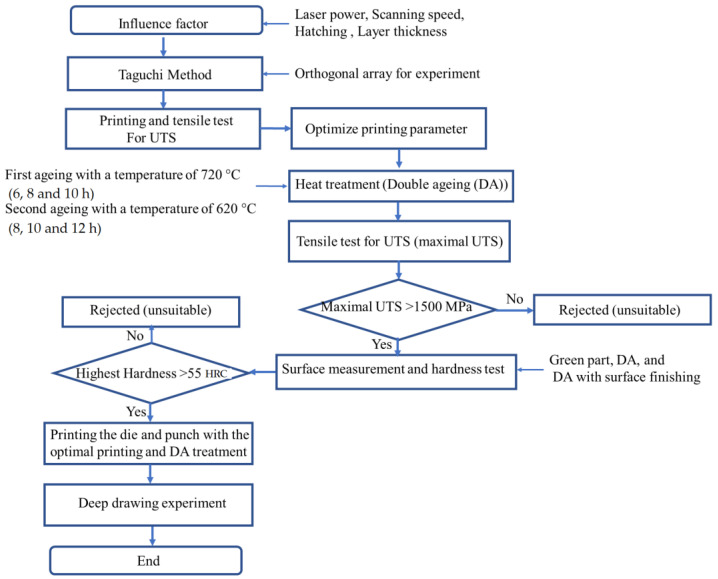
Research flow for this study.

**Figure 2 materials-16-04707-f002:**
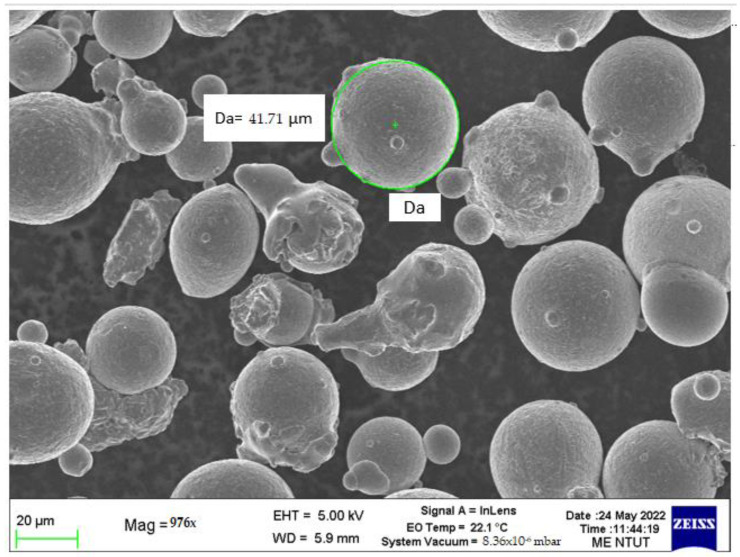
SEM image of IN 718 powder.

**Figure 3 materials-16-04707-f003:**
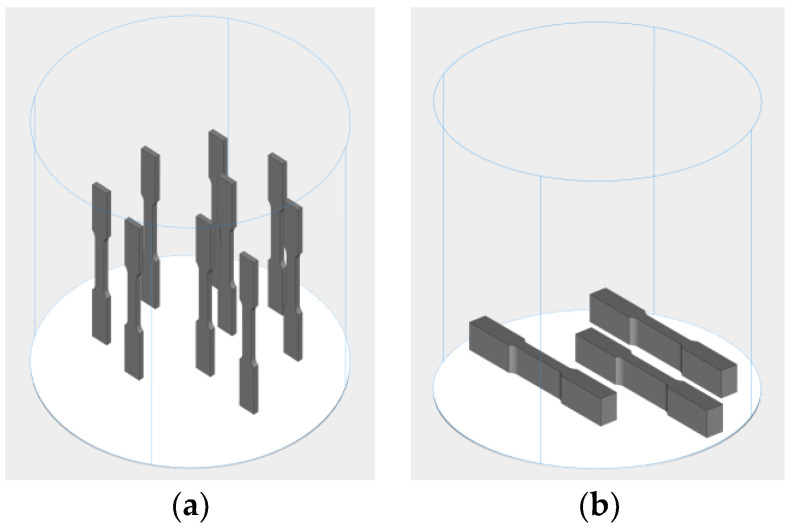
Tensile specimens using (**a**) vertical and (**b**) horizontal building directions.

**Figure 4 materials-16-04707-f004:**
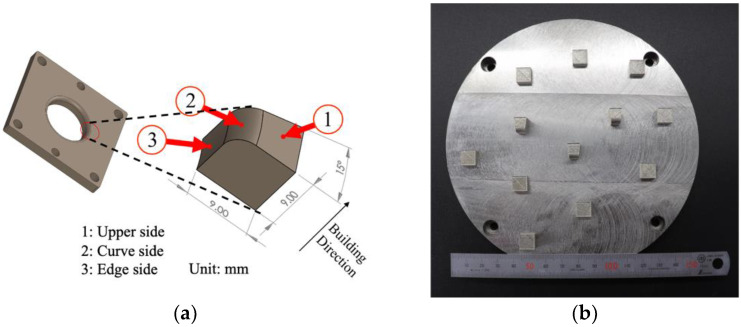
Hardness and surface roughness measurement points on part of die for deep drawing (**a**) and the as-built specimen (**b**).

**Figure 5 materials-16-04707-f005:**
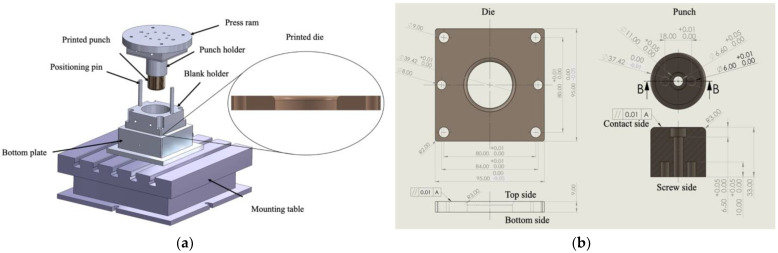
Schematic diagram of a deep drawing die set (**a**) and the dimensions of the die and punch (**b**).

**Figure 6 materials-16-04707-f006:**
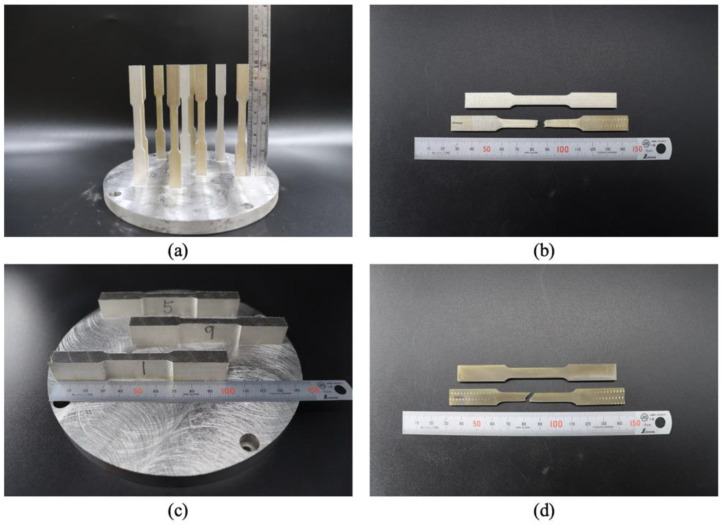
Photographs of the as-built specimens and the specimen after the tensile test: (**a**,**b**) are printed in the vertical building direction, and (**c**,**d**) are in the horizontal building direction.

**Figure 7 materials-16-04707-f007:**
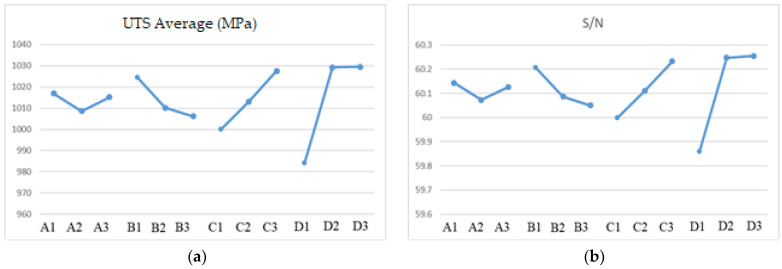
Factor response plot for (**a**) the average UTS and (**b**) the S/N ratio.

**Figure 8 materials-16-04707-f008:**
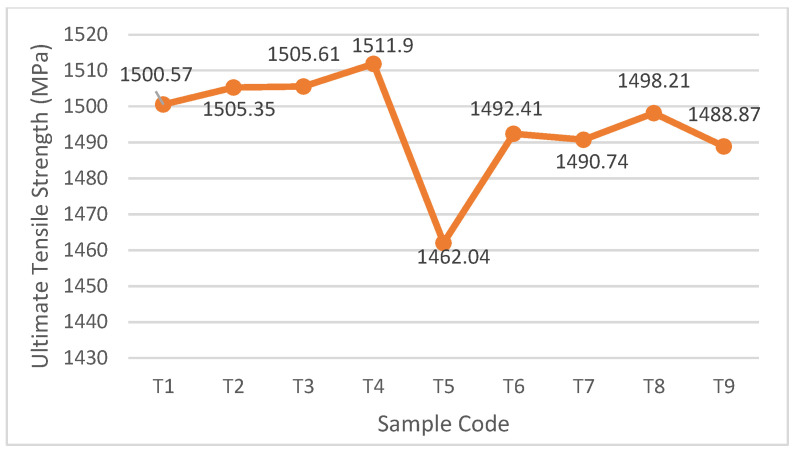
The maximum UTS for the tensile test for the specimen that is subject to DA treatment.

**Figure 9 materials-16-04707-f009:**
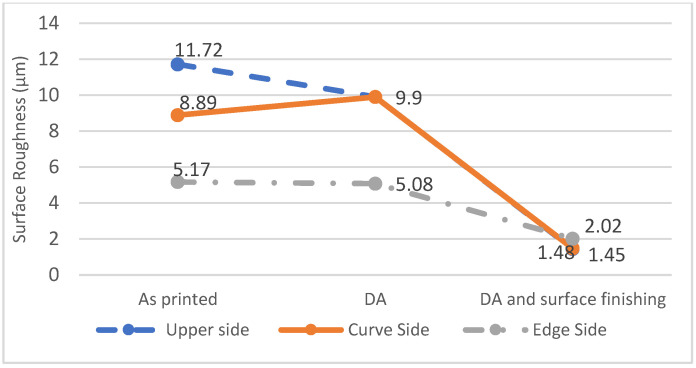
Surface roughness measurement for the as-built specimens, specimens that are subject to DA, and post-processed specimens (subject to DA and surface finishing).

**Figure 10 materials-16-04707-f010:**
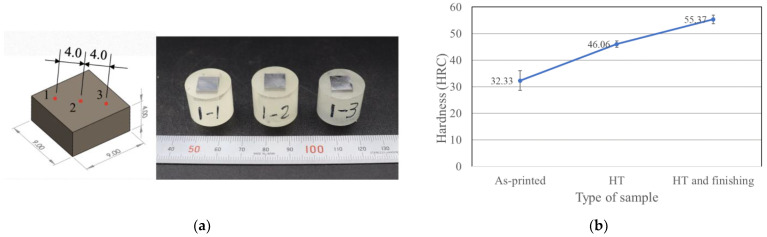
Rockwell hardness measurement: (**a**) the distance between each indentation point and (**b**) results for the as-built specimens, specimens that are subject to DA treatment, and post-processed specimens.

**Figure 11 materials-16-04707-f011:**
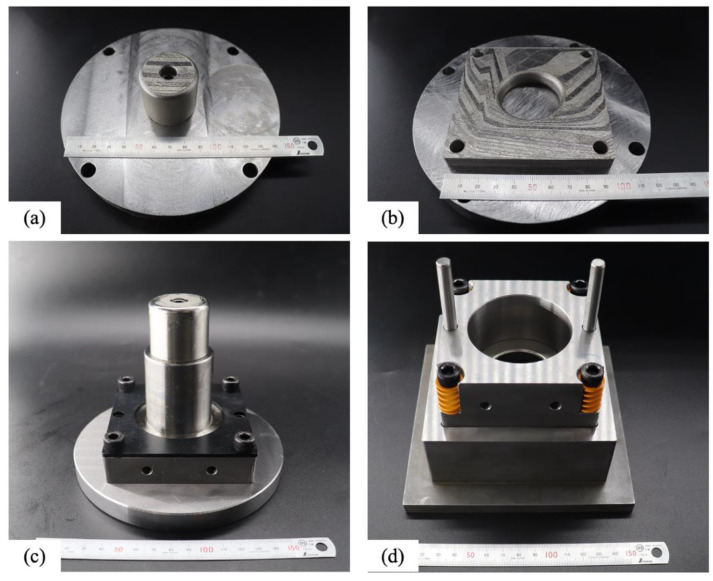
Photographs of the as-built punch (**a**) and die (**b**) on the platform and the assembled punch set (**c**) and die set (**d**).

**Figure 12 materials-16-04707-f012:**
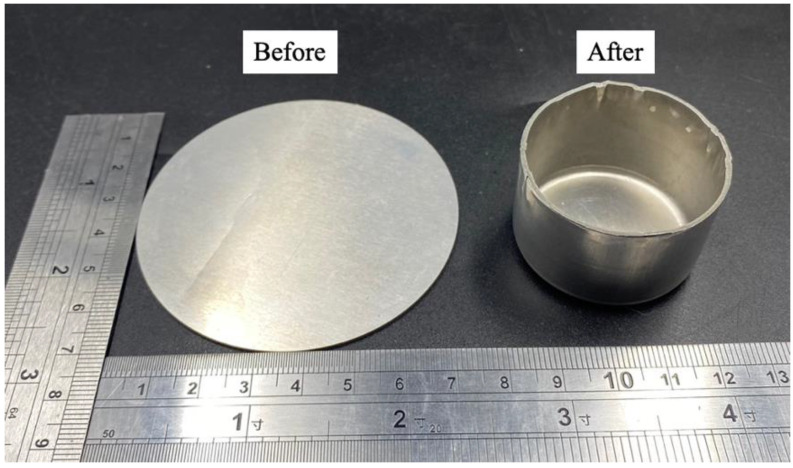
Photograph of the Al 6061 sheet (**left**) and the drawn round cup (**right**).

**Table 1 materials-16-04707-t001:** Printing parameters for IN 718 for different LPBF machines.

Laser Power (W)	Scanning Speed (mm/s)	Hatch Space (µm)	Layer Thickness (µm)	Ref.
180	600	105	35	[12]
200	700	-	60	[13]
350	600	80	40	[14]
200	800	105	30	[15]
190	800	90	30	[16]
180	600	105	30	[17]
170–370	500–1200	80–120	40	[18]

**Table 2 materials-16-04707-t002:** Printing parameters for the Taguchi Method.

Factor Code	Parameter/Level	1	2	3
A	Laser power (*P*)	180	190	200
B	Scanning speed (*V*)	600	700	800
C	Hatch space (*H*)	0.08	0.09	0.105
D	Layer thickness (*Lt*)	30	35	40

**Table 3 materials-16-04707-t003:** Orthogonal array for the experiment.

Sample Code (i)	Level				*P* (W)	*V* (mm/s)	*H* (mm)	*Lt* (µm)	VED (J/mm^3^)
	** *P* **	** *V* **	** *H* **	** *Lt* **					
1	1	1	1	1	180	600	0.08	30	125
2	1	2	2	2	180	700	0.09	35	81.63
3	1	3	3	3	180	800	0.105	40	53.57
4	2	1	2	3	190	600	0.09	40	87.96
5	2	2	3	1	190	700	0.105	30	86.17
6	2	3	1	2	190	800	0.08	35	84.82
7	3	1	3	2	200	600	0.105	35	90.7
8	3	2	1	3	200	700	0.08	40	89.28
9	3	3	2	1	200	800	0.09	30	92.59

**Table 4 materials-16-04707-t004:** Double ageing holding times for each experiment.

Sample Code	Double Ageing Holding Time (Hour)	
	First Ageing	Second Ageing
T1	6	8
T2	6	10
T3	6	12
T4	8	8
T5	8	10
T6	8	12
T7	10	8
T8	10	10
T9	10	12

**Table 5 materials-16-04707-t005:** *L*_9_ orthogonal result for the tensile test and the *S*/*N* ratio.

Sample Code (i)	Response			
	Vertical Direction		Horizontal Direction	
	Average UTS (MPa)	*S*/*N*	Average UTS (MPa)	*S*/*N*
1	942.72	59.87	1078.09	60.65
2	1028.75	60.24	1046.31	60.39
3	1037.4	60.32	1071.47	60.6
4	1033.01	60.28	1091.33	60.76
5	989.77	59.91	1073.11	60.61
6	1003.2	60.03	1048.41	60.41
7	1055.68	60.47	1056.63	60.45
8	1012.24	60.1	1056.8	60.48
9	977.78	59.8	1050.18	60.42

**Table 6 materials-16-04707-t006:** The optimal values for printing parameters for IN 718, as defined using the Taguchi method.

Sample Code (i)	Printing Parameter				Experimental Result UTS (MPa)	*S*/*N*	${VED}_{H}$ (J/mm^3^)
	*P*	*V*	*H*	*Lt*			
10	190	600	0.105	40	1122.88	60.58	75.4

## Data Availability

The data presented in this study are available upon request from the corresponding author.

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
