# Peer review of "Laser Powder Bed Fusion of Inconel 718 Tools for Cold Deep Drawing Applications: Optimization of Printing and Post-Processing Parameters"

_materials, 2023, doi:10.3390/ma16134707_

Round 1
Reviewer 1 Report
The authors presented an article on “Laser powder bed fusion of Inconel 718 tools for cold deep drawing applications: Optimization of printing and post-processing parameters”. The subject of the article falls within the scope of the journal "Materials". But the article was written in a sloppy way. There are many methodological errors. Thus, the article will be ready for publication after a major revision. Comments are listed below.
1) What is the novelty in this study? What is the difference between this study and similar studies in the literature? It should be explained.
2) What software was used for the Taguchi method. (For example, Minitab 16....) It must be specified.
3) The units of the parameters given in Table 1 should be added.
4) On page 5, line 168, "Table 1" should be "Table 2".
5) According to which standards were the hardness and surface roughness measurements made?
6) The Results section is devoid of discussion. It should be discussed by making comparisons with similar studies in the literature.
7) On page 6, line 201, "Figure 3(a)" should be "Figure 4(a)".
8) On page 7, line 205, "Figure 3" should be "Figure 4".
9) On page 7, line 210, "Figure 4(a)" should be "Figure 5(a)".
10) On page 7, line 217, "Figure 4(b)" should be "Figure 5(b)".
11) On page 7, line 222, "Figure 4" should be "Figure 5".
12) On page 7, line 227, "Figure 5" should be "Figure 6".
13) On page 8, line 234, "Figure 5" should be "Figure 6".
14) On page 8, line 238, "Figure 6" should be "Figure 7".
15) On page 8, line 239, "Figure 6(a)" should be "Figure 7(a)".
16) On page 9, line 247, "Figure 6(b)" should be "Figure 7(b)".
17) On page 9, line 257, "Figure 6" should be "Figure 7".
18) In Figure 6(a), the unit of the UTS average should be added.
19) On page 9, line 264, "Figure 7" should be "Figure 8".
20) The "x" axis header should be added in Figure 7.
21) In Figure 7, the reason for the sudden decrease in T5 UTS value should be explained.
22) On page 10, line 271, "Figure 7" should be "Figure 8".
23) On page 10, line 273, "Figure 8" should be "Figure 9".
24) On page 10, line 281, "Figure 8" should be "Figure 9".
25) In Figure 8, the unit of the y-axis should be added.
26) On page 10, line 283, "Figure 9" should be "Figure 10".
27) On page 10, line 283, "Figure 9(a)" should be "Figure 10(a)".
28) On page 10, line 285, "Figure 9(b)" should be "Figure 10(b)".
29) On page 11, line 294, "Figure 9" should be "Figure 10".
30) The "x" axis header should be added in Figure 9.
31) On page 11, line 297, "Figure 10(a)" should be "Figure 11(a)".
32) On page 11, line 301, "Figure 10(c-d)" should be "Figure 11(c-d)".
33) On page 11, line 313, "Figure 10" should be "Figure 11".
34) On page 12, line 316, "Figure 10" should be "Figure 12".
35) The article contains numerous typographic and language errors. It should be corrected.
36) The article should be rearranged by taking into account the journal writing rules and citation rules.
There is a reference problem. First, your reference list contains no article from the “Materials” journal. If your work is convenient for this journal's context, then there are many references from this journal. Secondly, cited sources should be primary ones. Namely, the indexed area shows the power of a paper and directly your paper's reliability. Please make regulations in this direction.
Author Response
First of all, I would like to express my sincere appreciation for the constructive comments, considerable time and significant efforts on my manuscript. Regarding the proposed modifications and points to be addressed, response is as attachment file.

Reviewer 2 Report
The author has well demonstrated the laser power bed fusion of Inconel 718 tools for cold deep drawing applications by optimizing printing and post-processing parameters. However, the author is suggested to take minor revision.
· The reviewer is interested to know on what basis the author has selected the process parameters of laser bed power fusion. Is there any effect of beam diameter, beam shape, pulse frequency, pulse duration on cold drawing performance of Inconel 718 tools or not?
· In Figure 1, there is some graphical error in the text used in block diagram. The author is suggested to take careful revision of entire manuscript to avoid such errors.
· In Equation (1), the author has used the abbreviation VED which is wrongly abbreviated in text, the author is suggested to take careful revision to avoid such errors.
· The reviewer is interested to know how the author has measure the surface roughness and microhardness as mentioned in section 3.3. Further, the author has not mentioned the unit of surface roughness which seems to be inappropriate from the reader point of view.
· The author is strongly advised to mention the limitation of the present study in the conclusion section.
· The author is strongly advised to mention the future scope of the present research work in the conclusion section.
Author Response
First of all, I would like to express my sincere appreciation for the constructive comments, considerable time and significant efforts on my manuscript. Regarding the proposed modifications and points to be addressed, response is as attachment.

Reviewer 3 Report
The paper is well-structured and nicely written. I commend the authors for this. They will need to address some of the minor and major points raised below. Minor:
- A small issue - the green text within the figure next to particle is not clearly visible. Please make it better.
- I was unable to easily find the definition of heat treatment (HT) parameter
- What is the temperature of cold drawing? room temperature? Authors need to clarify somewhere.
- I am not convinced UTS (via the Taguchi method) is sufficient. Is UTS the only thing that matters? I don't see stress strain curves from the tension experiments? What about ductility or failure strain? One may impact defects during printing and these defects may control the failure process and mechanical response under extreme environments.
Author Response

(The authors gave the same response as above.)

Round 2
Reviewer 1 Report
Necessary revisions were completed. This article can be accepted for publication in its final form.
Reviewer 3 Report
No more edits needed